# Data for wetlandscapes and their changes around the world

Navid Ghajarnia[1], Georgia Destouni[1], Josefin Thorslund[1], Zahra Kalantari[1], Imenne Åhlén[1], Jesús A. Anaya-Acevedo[2], Juan F. Blanco-Libreros[3], Sonia Borja[1], Sergey Chalov[4], Aleksandra Chalova[4], Kwok P. Chun[5], Nicola Clerici[6], Amanda Desormeaux[7], Bethany B. Garfield[8], Pierre Girard[9], Olga Gorelits[10], Amy Hansen[11], Fernando Jaramillo[1&12], Jerker Jarsjö[1], Adnane Labbaci[13], John Livsey[1], Giorgos Maneas[1&14], Kathryn McCurley[15], Sebastián Palomino-Ángel[16], Jan Pietroń[1&17], René Price[18], Victor H. Rivera-Monroy[19], Jorge Salgado[20], A. Britta K. Sannel[1], Samaneh Seifollahi-Aghmiuni[1], Ylva Sjöberg[21], Pavel Terskii[22], Guillaume Vigouroux[1], Lucia Licero-Villanueva[23], and David Zamora[24]

[1] Department of Physical Geography, Bolin Center for Climate Research, Stockholm University, SE-10691 Stockholm, Sweden.
[2] Facultad de Ingeniería, Universidad de Medellín, Carrera 87 30–65, 050026 Medellín, Colombia, S.A.
[3] Instituto de Biología, Facultad de Ciencias Exactas y Naturales, Universidad de Antioquia, Calle 70 No. 52-21, Medellín, Colombia
[4] Faculty of geography, Lomonosov Moscow State University, Moscow, Russian Federation; 119571, Moscow, Russia
[5] Department of Geography, Hong Kong Baptist University
[6] Department of Biology, Faculty of Natural Sciences and Mathematics, Universidad del Rosario, Bogotá D.C., Colombia
[7] School of Natural Resources and Environment, University of Florida, Gainesville, FL, USA
[8] Department of Geography and Anthropology, Louisiana State University, Baton Rouge, LA, 70803 USA
[9] Centro de Pesquisa do Pantanal, Cuiabá, Mato Grosso, Brazil
[10] Zubov State Oceanographic Institute, Moscow, Russian Federation
[11] Civil, Environmental and Architectural Engineering Department, University of Kansas, Lawrence, Kansas, USA
[12] Baltic Sea Centre
[13] Department of Geology,Faculty of Science of Agadir, Ibn Zohr University, Agadir, Morocco
[14] Navarino Environmnental Observatory, 24 001 Messinia, Greece
[15] Department of Soil and Water Sciences, University of Florida, Gainesville, Florida, USA
[16] Facultad de Ingeniería, Universidad de Medellín, Carrera 87 30–65, 050026 Medellín, Colombia, S.A.
[17] WSP Sverige AB, Ullevigatan 19, Gothenburg, 411 40, Sweden
[18] Department of Earth and Environment and Southeast Environmental Research Center, Florida International University, Miami, FL 33199, USA
[19] Department of Oceanography and Coastal Sciences, College of the Coast and Environment, Louisiana State University, Baton Rouge, LA, 70803 USA
[20] Departamento de Ciencias Biológicas, Universidad de Los Andes, Bogota Colombia; Universidad Católica de Colombia, Bogotá Colombia
[21] Department of Geosciences and natural resource management, CENPERM - Centre for Permafrost, University of Copenhagen, Copenhagen, Denmark
[22] Faculty of geography, Lomonosov Moscow State University, Moscow, Russian Federation; 119571, Moscow, Russia
[23] Master student in Landscape Ecology and Nature Conservation, University of Greifswald, Germany
[24] Civil and Agricultural Department, Universidad Nacional de Colombia – Bogotá, Colombia

**Abstract.** Geography and associated hydrological, hydroclimate and land use conditions and their changes determine the states and dynamics of wetlands and their ecosystem services. The influences of these controls are not limited to just the local scale of each individual wetland, but extend over larger landscape areas that integrate multiple wetlands and their total hydrological catchment – the wetlandscape. However, the data and knowledge of conditions and changes over entire wetlandscapes are still scarce, limiting the capacity to accurately understand and manage critical wetland ecosystems and their services under global change. We present a new Wetlandscape Change Information Database (WetCID), consisting of geographic, hydrological, hydroclimate and land use information and data for 27 wetlandscapes around the world. This combines survey-based local information with geographic shapefiles and gridded datasets of large-scale hydroclimate and land-use conditions and their changes over whole wetlandscapes. Temporally, WetCID contains 30-year time series of data for mean monthly precipitation and temperature, and annual land use conditions. The survey-based site information includes local knowledge on the wetlands, hydrology, hydroclimate and land uses within each wetlandscape, and on the availability and accessibility of associated local data. This novel database (available through PANGAEA https://doi.pangaea.de/10.1594/PANGAEA.907398; Ghajarnia et al., 2019) can support site assessments, cross-regional comparisons, and scenario analyses of the roles and impacts of land use, hydroclimatic and wetland conditions and changes on whole-wetlandscape functions and ecosystem services.

## 1 Introduction

Wetlands contribute more than 20% of the total value of global ecosystem services (Costanza et al., 2014), while covering only a small percentage (4-9%) of global land surface (Morganti et al., 2019; Zedler and Kercher, 2005; Mitsch and Gosselink, 2000). Wetlands are associated with a diverse range of functions such as water quality remediation (e.g., Chalov et al., 2017; Quin et al., 2015), regulation of soil moisture and groundwater replenishment (e.g., Ameli and Creed, 2019; Golden et al., 2017), flood control (e.g., Quin and Destouni, 2018; Acreman and Holden, 2013), and biodiversity conservation (e.g., Cohen et al., 2016; Mitchell et al., 2008). Through these functions, wetlands can support regional sustainability (Seifollahi-Aghmiuni et al., 2019) but are also one of the most vulnerable ecosystems globally (Golden et al., 2017). For instance, human land and/or water use developments (Destouni et al., 2013; Jaramillo and Destouni, 2015; Maneas et al., 2019) in combination with climate variability and change (Orth and Destouni, 2018; Seneviratne et al., 2006) affect large-scale water fluxes with impacts on wetland functions and ecosystem services. These impacts extend over coupled systems of multiple wetlands and the associated total hydrological catchment that integrates these, referred to as a wetlandscape (Thorslund et al., 2017), with even well-intended actions towards various sustainable development goals potentially affecting wetland functions and services in different directions (Jaramillo et al., 2019). As a consequence of various change impacts, wetland areas are now suffering rapid and continued decline in different regions worldwide (Davidson et al., 2018; Davidson, 2014).

The scale mismatch between the existing large-scale studies of various landscape changes and the still mostly local wetland impact studies (Thorslund et al., 2017) creates an urgent need for comprehensive, science-based assessment of the interactions between large-scale drivers of change and large-scale wetland systems (Ameli and Creed, 2019; Creed et al., 2017). Adopting a wetlandscape perspective involves moving away from the individual wetland scale to consider the large-scale functioning of the hydrologically coupled system of multiple wetlands and their surrounding landscape. Assessments at these larger scales are needed to enable the formulation of scientific evidence-based guidance and strategies to protect wetlands under global change (Thorslund et al., 2018; Ameli and Creed, 2019). The conceptual framework on wetlandscapes was developed over 30 years ago, by Preston and Bedford (1988), but the dynamics and impacts of many large-scale drivers or functions on wetlandscape scales remain still largely uninvestigated and unknown, with the interactions between large-scale hydroclimatic variability and change and wetland dynamics still being largely underexplored at wetlandscape scale (Thorslund et al., 2017). The combination of high wetland vulnerability and rapid large-scale changes subject to major knowledge and data gaps highlights the need to synthesize and create datasets available for evaluating change effects and feedbacks on the scales of whole wetlandscapes.

To address this need and support large-scale studies of whole wetlandscapes in and across different parts of the
world, we have created a novel database named as the Wetlandscape Change Information Database (WetCID), for
27 wetlandscapes around the world and their associated geographical, wetland, hydrology, hydroclimate, and land use
conditions. WetCID consists of a survey-based collection of local information and data, combined with compilation
and synthesis of gridded large-scale datasets for a range of relevant hydroclimatic and land use variables.

The remainder of this paper is structured as follows: In section 2, we describe the methodology used in collecting,
processing, and summarizing different datasets. In section 3, we present WetCID summaries and sample figures and
maps from different components of the underlying datasets, in order to exemplify and highligt the potential of new
insights that can be gained from using this database, as well as its limitations. In section 4, we discuss data availability
and the format and structure of different files in WetCID. Based on the findings, we present some conclusions in
section 5.

## 2 Methods
### 2.1 Data acquisition
In compilation of WetCID for the 27 wetlandscapes, we employed three sources of primary data. These were: (1)
local site survey data, depicting general characteristics of each wetlandscape (catchment) and its geographical
characteristics (including shapefiles for its spatial extent) and associated hydrological, climate, and land use conditions
and their observed/perceived changes; (2) gridded historical data time series of monthly precipitation and temperature
from Climate Research Unit Time-Series (CRU_TS) version 4.02 (Harris et al., 2014); and (3) historical data of annual
land cover and its changes from the NOAA-HYDE dataset provided by NOAA's National Climate Data Center (Jain
et al., 2013; Meiyappan and Jain, 2012).

The survey for local site data (1) was given to researchers within the Global Wetland Ecohydrological Network
(GWEN) (www.gwennetwork.se). The GWEN researchers responding to the survey specified the relevant
wetlandscape extent (total hydrological catchment with wetlands) and provided boundaries in GIS format for the 27
wetlandscapes, located as shown in Figure 1. Information and data of all three types (local survey-based, hydroclimate,
land use) were collected and synthesized for each of these wetlandscapes from all three sources (1)-(3). In addition to
the local survey information, data on hydroclimate and land use variables were thus also compiled from the global
datasets in both gridded and aggregated form for each wetlandscape, as described further in the following.

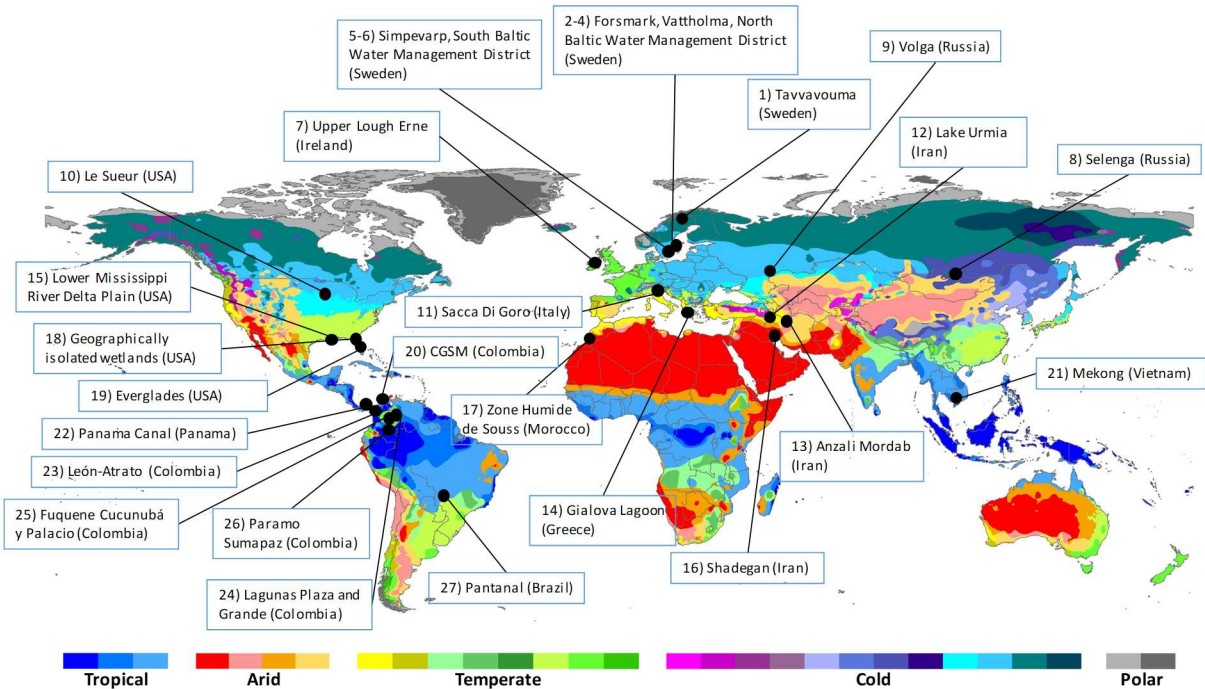

**Figure 1.** Geographical distribution of the 27 wetlandscape sites included in WetCID. The background map shows the Köppen-Geiger climate classification system (as updated by Peel et al., 2007), with the number of wetlandscapes extended from those included in similar GWEN-site mapping by Thorslund et al. (2017). The site numbering is in order of latitude from north to south, covering a latitude range from 70°N to 25°S.

## 2.2 Site information surveys

A questionnaire for collecting local site knowledge and information on the availability and accessibility of local data was developed during a GWEN workshop held in Santa Marta, Colombia, on April 24-28, 2018. The questionnaire was sent out by email after the workshop to all participating GWEN researchers. The researchers responding to it related their answers to a specific wetlandscape in which they had active research.

The questionnaire comprised two main parts. Part 1 contained general questions about the geography, climate, hydrology, and wetland-relevant human activities and changes in the wetlandscapes. Part 2 focused on the availability and accessibility of local site data, structured into 'Hydroclimate', 'Land use', and 'Other' data (see templates in the database files for a full outline of the questionnaire). The collective knowledge obtained on conditions and changes in the 27 wetlandscapes and on data availability-accessibility is summarized in section 3.1.

To complement this local knowledge and information basis, we further extracted and synthesized data for the 27 wetlandscapes from relevant global hydroclimate and land use datasets as described below.

## 2.3 Hydroclimate data

The temperature and precipitation data taken from the CRU_TS4.02 global datasets (Harris et al., 2014) covered a 30-year period (1981-2010), to be consistent with the time span of existing global land use change data. CRU_TS4.02 provides hydroclimate data with spatial resolution of 0.5° × 0.5° and at monthly temporal scale. In preparing temperature and precipitation datasets for each wetlandscape, the gridded data within the area of the wetlandscape were extracted from the global datasets and also spatially aggregated over that area, based on area-weighted averaging over the grid cells covered by the shapefile of each wetlandscape (catchment). This provided wetlandscape-specific data time series for each variable at each grid cell and aggregated over the whole wetlandscape. To facilitate analyses

at different spatial resolutions, both the gridded and the aggregated time series were included in WetCID for each of
the 27 wetlandscapes.

In addition to the gridded and aggregated data time series, period-specific temperature and precipitation changes were
also calculated  for each wetlandscape, by dividing the total 30-year time span of the collected data into the two 15-
148    year periods 1981-1995 (Per1) and 1996-2010 (Per2). Such period-specific change quantification can facilitate
relatively simple and straightforward analysis of how these hydroclimatic changes correlate with and may have driven
other wetlandscape changes (e.g., in runoff, evapotranspiration, wetland area) between the same time periods
(Destouni et al., 2013; Jaramillo and Destouni, 2014, 2015). Absolute and relative (%) changes between these periods
($AbsChng$ and $RelChng$, respectively) were calculated from the mean annual values of temperature and precipitation
during Per1 and Per2, as:

$$AbsChng = \overline{Var_{Per2}} - \overline{Var_{Per1}} \qquad (1)$$

$$RelChng = \frac{\overline{Var_{Per2}} - \overline{Var_{Per1}}}{\overline{Var_{Per1}}} \times 100 \qquad (2)$$

where $\overline{Var_{Per1}}$ and $\overline{Var_{Per2}}$ are average temperature (in C°) or precipitation (in mm/yr) over Per1 (1981-1995) and
Per2 (1996-2010), respectively. Eq. (1) was applied to both temperature and precipitation data, to calculate their
absolute changes in each wetlandscape, while Eq. (2) was only applied to precipitation data, to calculate the
corresponding percentage change in precipitation.
### 2.4 Land use data
The NOAA-HYDE dataset was used to estimate land uses and their changes in each wetlandscape. NOAA-HYDE
estimates annual changes in land cover area over the global land mass, starting from a base map for year 1765. The
estimations follow a predefined pathway, determined by relevant land use/management datasets (cropland,
pastureland, urbanization, timber harvesting), to obtain forest area distributions close to satellite-based estimates of
forests in recent years (Meiyappan and Jain, 2012). NOAA-HYDE data cover the period 1770-2010 with yearly
temporal resolution and spatial resolution of 0.5° × 0.5°, from which data for the period 1981-2010 were used for the
development of this database, in consistency with the hydroclimate data period described above.

The NOAA-HYDE land cover maps show the percentage of grid cell area containing 28 different land cover types
(see Table A1 in Appendix A). In this study, we reclassified these 28 land cover types into 10 distinct land covers:
urban, shrubland, grassland, pastureland, cropland, forest, water, desert, tundra, and savannah, by combining similar
land cover classes (see Table A1). As done for the hydroclimate data, the gridded land use data were also spatially
aggregated over each wetlandscape based on the area-weighted averaging method (with weights of specific land-cover
area in each grid cell relative to total wetlandscape area). This provided a wetlandscape-specific data time series of
annual land use/cover, for each of the reclassified 10 land cover types. The final WetCID files comprised gridded time
series data on absolute grid cell area (in km$^2$) covered by each land cover type, time series data on percentage of grid
cell area covered by each land cover type, and aggregated absolute and percentage time series data for each
wetlandscape area.

In analogy with the hydroclimatic changes, period-specific change quantification can facilitate relatively simple and
straightforward analysis of how different types of land use changes between time periods correlate with and may have
driven associated wetlandscape changes (Destouni et al., 2013; Jaramillo and Destouni, 2015). Eq. (1) was therefore
also used to calculate absolute change in the area of each land cover type (km$^2$) within each wetlandscape between
Per1 (1981-1995) and Per2 (1996-2010). In the land use case, $\overline{Var_{Per1}}$ and $\overline{Var_{Per2}}$ represent annual average area
covered by a land cover type within each wetlandscape during Per1 and Per2, respectively. Furthermore, the
corresponding change in relative land cover area ($ChngRel$ in %-points of total wetlandscape area) was calculated as:

$$ChngRel = \frac{\overline{Var_{Per2}} - \overline{Var_{Per1}}}{Area_C} \times 100 \qquad (3)$$

where $Area_C$ is the total wetlandscape (catchment) area in km$^2$ and $\overline{Var_{Per1}}$ and $\overline{Var_{Per2}}$ are the annual average areas
covered by each land cover type in the wetlandscape during Per1 and Per2, respectively.

**3 Results**
**3.1 Site information surveys**
Table 1 summarizes some general geographical, climate, and wetland type information provided by GWEN
researchers in the survey information forms. Each site represents either an individual wetland or a wetlandscape (e.g.,
a catchment) including multiple wetlands. The country, main climate zone and wetland area relative to total
wetlandscape (catchment) area are also given for each site in Table 1. Moreover, a summary of the availability-
accessibility of local data on the wetlands, hydrology, climate, and land uses, as well as the wetlandscape (catchment)
area in each of the 27 wetlandscapes is also shown in Figure 2. The variables of evapotranspiration and soil moisture
were revealed as having large data gaps (red color in Figure 2), indicating an overall need to use other data sources
(e.g., gridded global data products) for quantifying these variables and associated processes. Figure 2 also highlights
the variability in data availability and open accessibility among the sites. For instance, no open data sources have been
reported for the considered variables in the arid subtropical sites 13, 16, and 17, whereas open data sources have been
reported for most variables in the cold Swedish sites 4 and 6, and the American subtropical sites 15 and 19.

The synthesized survey dataset also contains information about different types of wetland, hydroclimatic and/or land
use changes observed/perceived to have occurred in the 27 investigated wetlandscapes (Figure 3). Substantial changes
are reported for most of these wetlandscapes, but a few sites have no known changes (e.g., in the arid Morroccan site
17) or have important knowledge gaps regarding changes (e.g., in the cold Swedish sites 2 and 5, even though
availability to at least some data is relatively good there). The information on local data availability-accessibility
(Figure 2) and observed/perceived change occurrence (Figure 3) summarised and structured in WetCID can guide
further study directions, and support identification of key needs for complementary new local data and/or use of
additional large-scale (regional-global) gridded data. Furthermore, the wetlandscapes of WetCID are located in
different regions of the world, with seven sites in Northern Europe (sites 1-7), seven in the Amazon and Caribbean
region (sites 20 and 23-27), four in North America (sites 10, 15, 18, and 19), three in the Middle East (sites 12, 13,
and 16), two in the Mediterranean region (sites 11 and 14), two in Siberia (sites 8 and 9), and two more in other parts
of the world (Northern Africa and East Asia). As such, regional patterns and characteristics can be identified, and
regional strategies developed, e.g., to enhance availability of data and information, and determine further research
needed to bridge region-specific knowledge gaps and decide on relevant management plans for each region's wetland
ecosystems. Such regional characterizations and assessments can be initialized with the current version of WetCID
and further updated as more data for already included and possible additional regional wetlandscapes become available
in future database versions.

**Table 1.** General geographic, climate, and wetland type information for the 27 investigated wetlandscapes in WetCID. The data and information are based on
survey responses by researchers with active research (on various topics) at each wetlandscape site.

| Site No. | Site name | Country | Classification | Climate zone | Wetland type | Area of wetlands relative to total catchment/wet-landscape area (%) |
|---|---|---|---|---|---|---|
| 1 | Tavvavouma | Sweden | Wetlandscape | Subarctic | Peat plateau/thermokarst lake complex | 2.8 |
| 2 | Forsmark | Sweden | Wetlandscape | Humid continental (cold summer) | Bogs, fens, marshes, (shallow lakes) | 0.01 |
| 3 | Vattholma | Sweden | Wetlandscape | Humid continental (cold summer) | Bog, Fen, Riparian | - |
| 4 | North Baltic WMD | Sweden | Wetlandscape | Humid continental (cold summer) | Multiple | 100 |
| 5 | Simpevarp | Sweden | Wetlandscape | Humid continental (cold summer) | Bogs, fens | 0.01 |
| 6 | South Baltic WMD | Sweden | Wetlandscape | Humid continental (cold summer) | Multiple | 100 |
| 7 | Upper Lough Erne | Ireland | Individual wetland | Cold (dry winter, cold summer) | Flood plain/shallow lakes | 22 |
| 8 | Selenga | Russia | Wetlandscape | Cold (dry winter, cold summer) | Marshes (Riverine, Palustrine) | 0.13 |
| 9 | Volga | Russia | Wetlandscape | Cold (dry winter, cold summer) | Marshes (Riverine, Palustrine) | 1.0 |
| 10 | Le Sueur | USA | Wetlandscape | Temperate | isolated, fluvial/riparian, lakes/ponds, marshes, forest/shrubs, constructed | 100 |
| 11 | Sacca Di Goro | Italy | Individual wetland | Cold-summer Mediterranean | Shallow saltwater coastal lagoon | 4.2 |
| 12 | Lake Urmia | Iran | Individual wetland | Continental | Lake | 8.8 |
| 13 | Anzali Mordab | Iran | Individual wetland | Caspian or Hyrcanian climate | Inland and Marine/Coastal wetland | 4.0 |
| 14 | Gialova Lagoon | Greece | Individual wetland | Hot-summer Mediterranean | Coastal wetland | 13 |
| 15 | Lower Mississippi River Delta Plain | USA | Wetlandscape | Humid Subtropical | Riverine, Marine, Estuarine, Lacustrine | 3.5 |
| 16 | Shadegan | Iran | Individual wetland | Warm desert | Palustrine, Estuarine, Marin | 31 |
| 17 | Zone Humide de Souss | Morocco | Individual wetland | Mediterranean semi-arid | marine and coastal | 0.01 |
| 18 | Geographically isolated wetlands | USA | Wetlandscape | Humid subtropical | Freshwater marshes and swamps | 100 |
| 19 | Everglades | USA | Individual wetland | Tropical to Subtropical | Freshwater wetland, coastal wetland | 32 |
| 20 | CGSM | Colombia | Individual wetland | Tropical | Estuarine | - |
| 21 | Mekong Delta | Vietnam | Wetlandscape | Tropical Monsoon | Marine | 5.0 |
| 22 | Panama Canal | Panama | Wetlandscape | Tropical/Central America | River Chagres, Lake | 100 |
| 23 | León-Atrato | Colombia | Wetlandscape | Tropical rainforest | Marshes and Swamps | 17 |
| 24 | Lagunas Plaza and Grande | Colombia | Wetlandscape | Extremely cold and very dry | Glacial Lake | 4.4 |
| 25 | Fúquene, Cucunubá y Palacio | Colombia | Individual wetland | Cold and very dry | Natural shallow lake | 1.7 |
| 26 | Paramo Sumapaz | Colombia | Wetlandscape | Tropical | High altitude wetland | 46 |
| 27 | Pantanal | Brazil | Wetlandscape | Tropical savanna with dry-winter | Periodically inundated savanna | 27 |

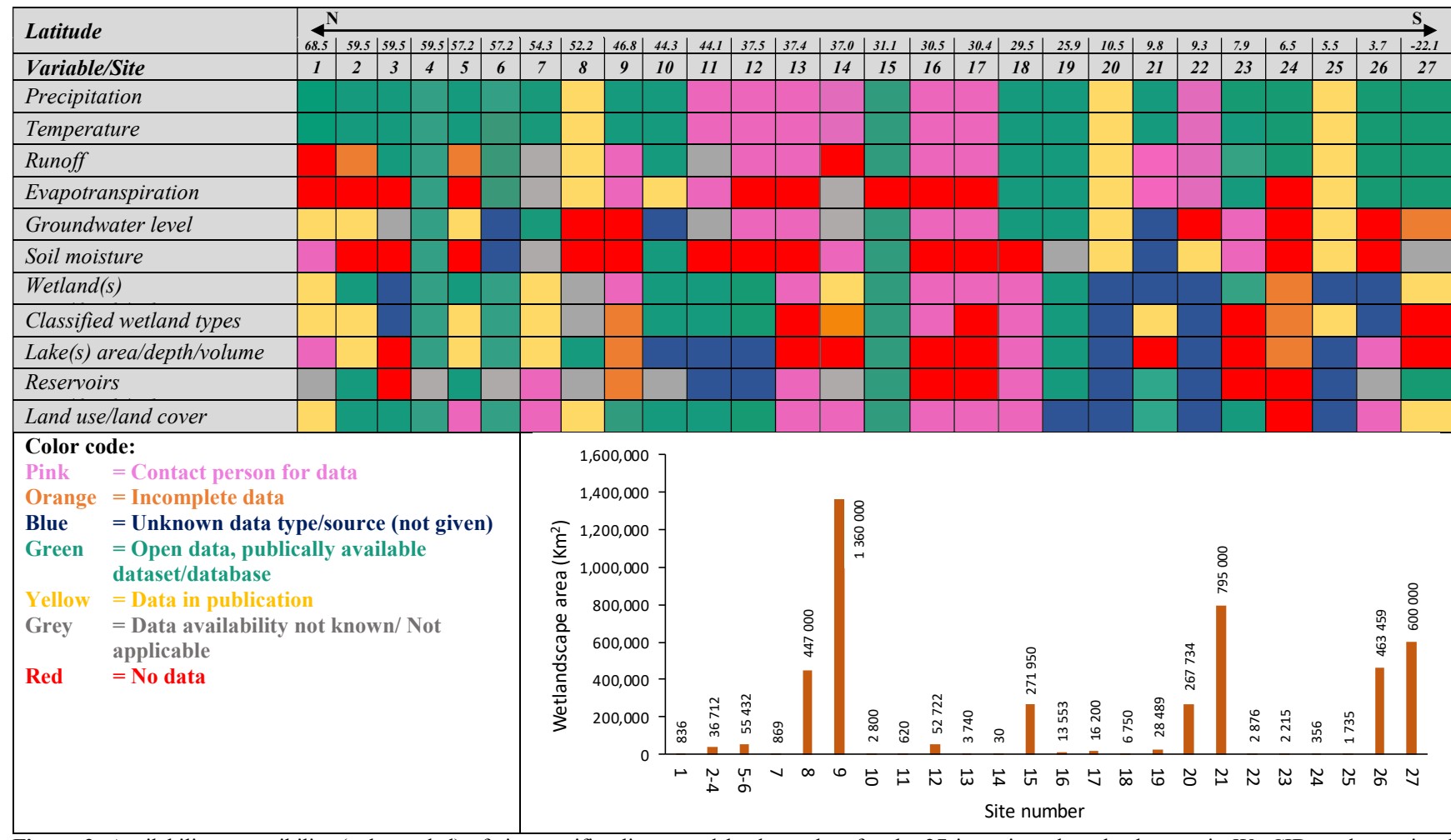

Figure 2. Availability-accessibility (color-coded) of site-specific climate and land use data for the 27 investigated wetlandscapes in WetCID, and associated wetlandscape area for each site (lower right diagram). The data availability-accessibility classification (color codes) is based on the survey responses by researchers with active research (on various topics) at each wetlandscape site.

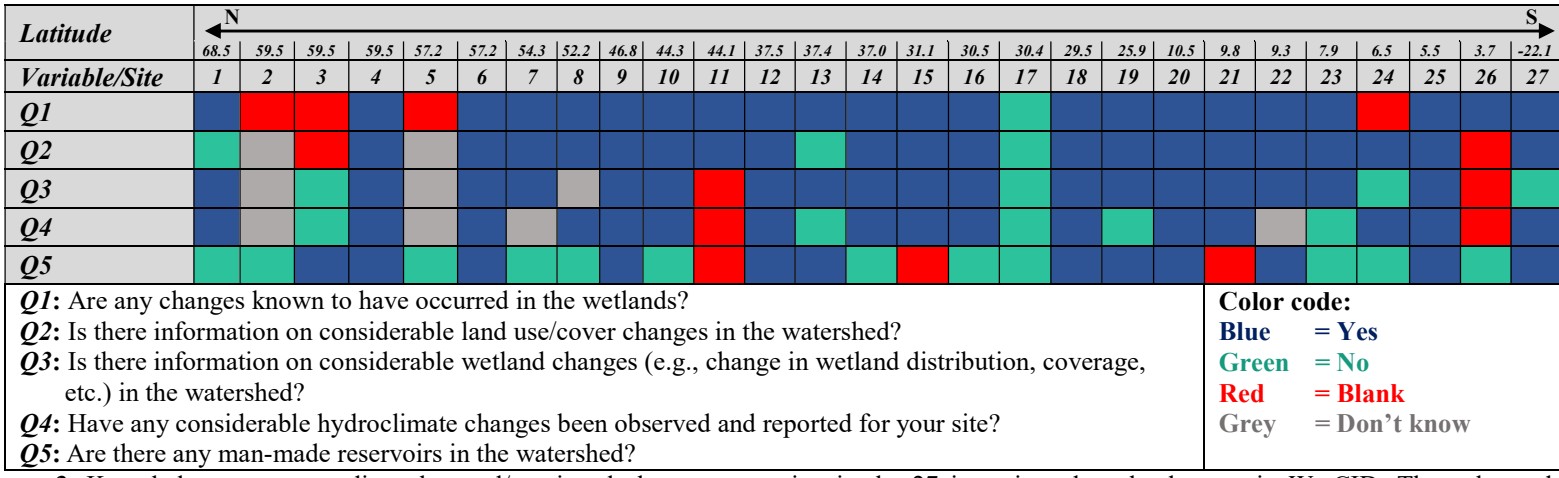

| Latitude | N ← | | | | | | | | | | | | | | | | | | | | | | | | | → S |
|---|---|---|---|---|---|---|---|---|---|---|---|---|---|---|---|---|---|---|---|---|---|---|---|---|---|---|---|
| | 68.5 | 59.5 | 59.5 | 59.5 | 57.2 | 57.2 | 54.3 | 52.2 | 46.8 | 44.3 | 44.1 | 37.5 | 37.4 | 37.0 | 31.1 | 30.5 | 30.4 | 29.5 | 25.9 | 10.5 | 9.8 | 9.3 | 7.9 | 6.5 | 5.5 | 3.7 | -22.1 |
| Variable/Site | 1 | 2 | 3 | 4 | 5 | 6 | 7 | 8 | 9 | 10 | 11 | 12 | 13 | 14 | 15 | 16 | 17 | 18 | 19 | 20 | 21 | 22 | 23 | 24 | 25 | 26 | 27 |
| Q1 | | | | | | | | | | | | | | | | | | | | | | | | | | | |
| Q2 | | | | | | | | | | | | | | | | | | | | | | | | | | | |
| Q3 | | | | | | | | | | | | | | | | | | | | | | | | | | | |
| Q4 | | | | | | | | | | | | | | | | | | | | | | | | | | | |
| Q5 | | | | | | | | | | | | | | | | | | | | | | | | | | | |

**Q1:** Are any changes known to have occurred in the wetlands?
**Q2:** Is there information on considerable land use/cover changes in the watershed?
**Q3:** Is there information on considerable wetland changes (e.g., change in wetland distribution, coverage, etc.) in the watershed?
**Q4:** Have any considerable hydroclimate changes been observed and reported for your site?
**Q5:** Are there any man-made reservoirs in the watershed?

Color code:
Blue = Yes
Green = No
Red = Blank
Grey = Don't know

**Figure 3.** Knowledge status regarding observed/percieved changes occurring in the 27 investigated wetlandscapes in WetCID. The color-coded status classification is based on survey responses by researchers with active research (on various topics) at each wetlandscape site.

### 3.2 Hydroclimatic data

Data for long-term average temperature and precipitation conditions, and changes in these between Per1 (1981-1995) and Per2 (1996-2010) at the 27 wetlandscape sites are presented in Figure 4. The horizontal axis in the diagrams shows the wetlandscape site numbers in order of their latitude from north to south, covering the latitude range from 70°N to 25°S. The increase in average temperature and precipitation with decreasing latitude (Figure 4a, 4b) illustrates that the wetlandscapes also cover a wide range of hydroclimate conditions, from low to high temperature and precipitation values (see also Figure 1). Temperature has increased over almost all wetlandscapes, and considerably more so in the more northern and colder areas than in the warmer areas around and south of the equator (Figures 4a-b). In contrast, precipitation changes are relatively small, varying around zero, in the more northern, colder as well as drier areas, while precipitation has mostly increased in the warmer and also wetter areas around and south of the equator (Figures 4c-4e). Overall, the changes in mean annual temperature range from zero to +1°C while the changes in precipitation range from -70 mm/yr to +170 mm/yr, with the Iranian site 12 (Lake Urmia catchment) exhibiting the greatest increase in temperature (+1°C) and the greatest relative decrease in precipitation (-17%).

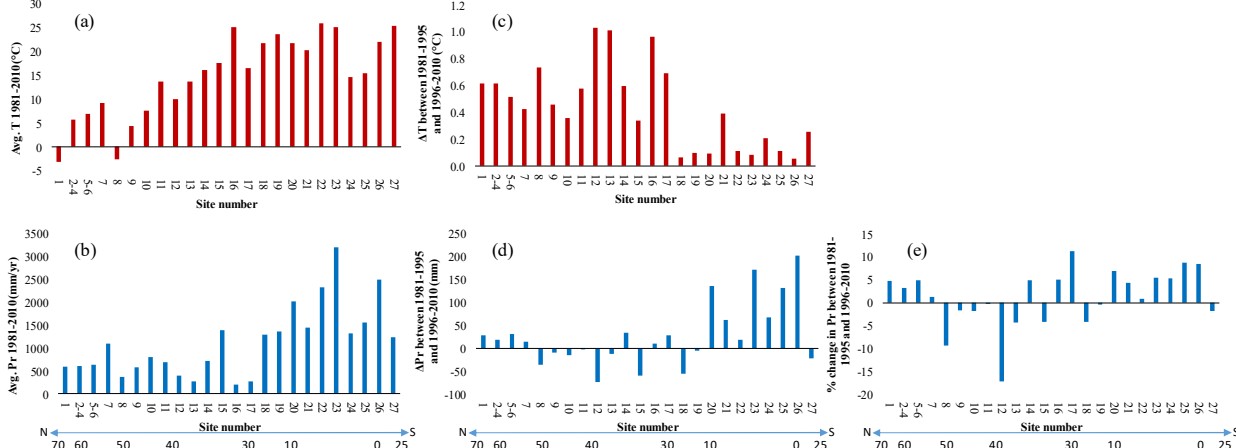

**Figure 4.** Overview of hydroclimate conditions and their changes in the 27 wetlandscapes. Long-term average (1981-2010) (a) temperature and (b) precipitation. Absolute change between Per1 (1981-1995) and Per2 (1996-2010) in (c) mean annual temperature and (d) mean annual precipitation. (e) Relative change in precipitation. The horizontal axis shows the numbering of the 27 wetlandscapes, sorted in order of their latitude from North to South.

Figures 5 and 6 exemplify gridded variability and change data for temperature and precipitation over the Volga (no. 9) and the León-Atrato (no. 23) wetlandscapes. The data times series of wetlandscape-aggregated annual average temperature and precipitation in these wetlandscapes (Figure 5) exemplify such data prepared and included in WetCID for all 27 wetlandscapes. These two wetlandscapes were chosen for data exemplification because they represent different hydroclimatic conditions, with Volga being cold and dry while León-Atrato is warm and wet (Figure 5), as well as have different sizes with Volga being the largest (1,360,000 km²) and León-Atrato (2,344 km²) one of the smallest studied wetlandscapes. The data for these examples (Figure 5) are consistent with corresponding data implications across the different wetlandscapes over the world (Figure 4) in indicating an overall positive (warmer-wetter) spatial correlation between long-term average temperature and precipitation. Temporally, however, the recent changes in these variables imply a negative correlation (towards warmer and mostly drier conditions) for the Volga wetlandscape (Figure 6, left) as for several other northern wetlandscapes in WetCID (Figure 4). In contrast, a positive correlation (towards mostly warmer and wetter conditions) is implied by the recent temporal changes in the León-Atrato wetlandscape (Figure 6, right) as one of the most sourthern wetlandscapes in WetCID (Figure 4). Such spatiotemporal sign shifts and dipole emergence in temperature-precipitation correlations have been noted in other recent studies of long-term variations and short-term changes of hydroclimate over Europe (Charpentier Ljungqvist et al., 2019). WetCID can facilitate further studies of these correlation conditions for and across the different wetlandscapes around the world.

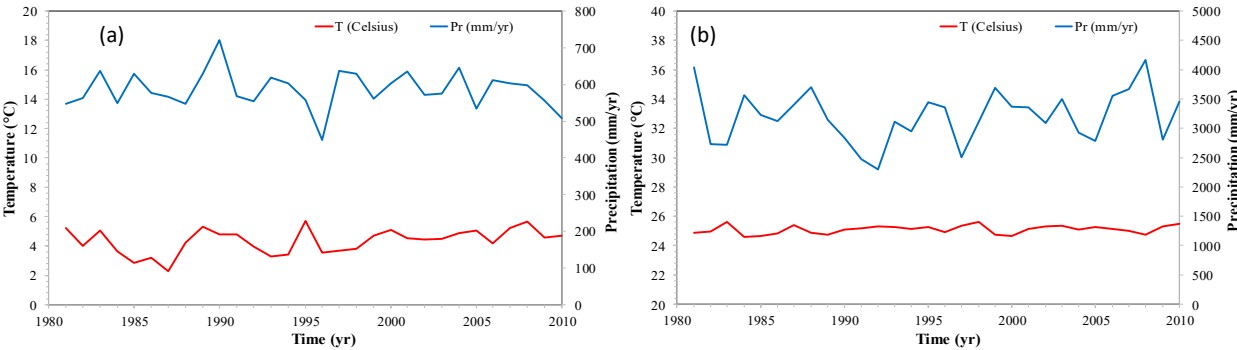

**Figure 5.** Variability in wetlandscape-aggregated annual average temperature and precipitation for the examples of
the (a) Volga and (b) León-Atrato wetlandscapes.

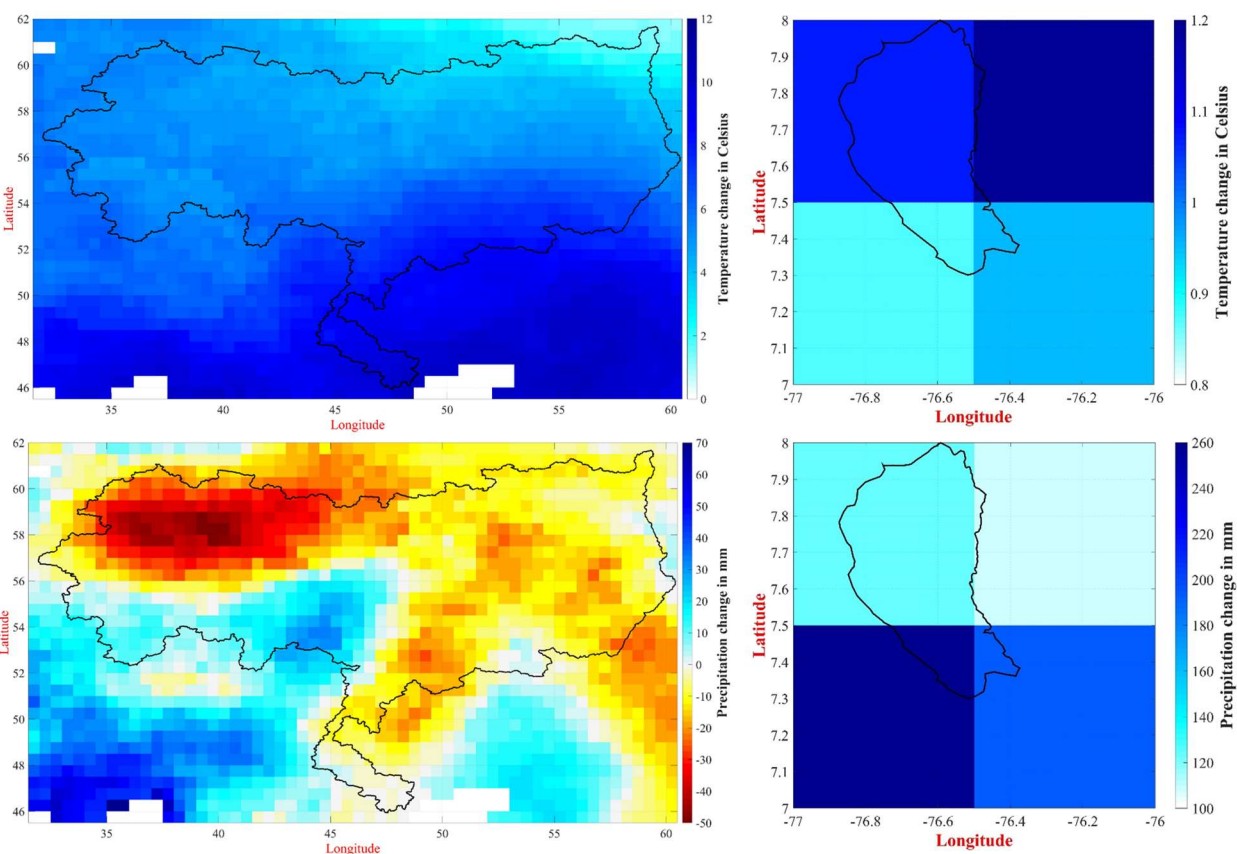

**Figure 6.** Maps showing gridded absolute change in (upper diagrams) temperature and (lower diagrams) precipitation
for the examples of the (left) Volga and (right) León-Atrato wetlandscapes. Absolute change values have been
calculated by applying Eq. (1) on each grid cell within a wetlandscape.

The data for the Volga and León-Atrato examples also emphasize that wetlandscapes can have very different area
extents (spatial scales), with potentially important implications for the spatial resolution (Figure 6) and related
usefulness of data provided in WetCID. For example, the Volga wetlandscape includes 982 grid cells with complete
or partial coverage in the hydroclimate datasets, while the León-Atrato wetlandscape only includes 4 such grid cells.
Most of the available global datasets from climate and earth system models have coarser spatial resolution than the
size of most individual wetlands. Thus, model data for individual wetlands are subject to high uncertainty, whereas
data aggregated over whole wetlandscapes have greater potential for accuracy (Bring et al., 2015), highlighting the
need for considering the whole-wetlandscape scales in assessments of how wetland systems interact with hydroclimate
and land use changes.

## 3.3 Land use data

The aggregated and gridded land use data in WetCID can also be used for different types of whole-wetlandscape analyses. Figure 7 summarises the data for long-term average relative area of each land cover type (% of total wetlandscape area), and associated absolute area changes (km²) and changes in relative area coverage (%-points of total wetlandscape area) for different land cover types across the 27 wetlandscapes. The data reveal, for example, the high percentage of forest area in wetlandscapes at high latitudes and in the tropics, while relative cropland area increases towards the temperate regions (Figure 7, left). Figure 7 also summarises the different types of land cover transformations, for example from: 'forest' into 'cropland and pastureland' in the tropical Mekong wetlandscape 21; 'pastureland' into 'grassland' in the temperate Irish wetlandscape 7 and into 'cropland' in the borderline cold-dry Iranian wetlandscape of the dramatically shrinking Lake Urmia 12 (Khazaei et al., 2019); 'shrubland' into 'cropland' in the borderline temperate Iranian wetlandscape 13; 'cropland' into 'shrubland' in the warm temperate Greek wetlandscape 14.

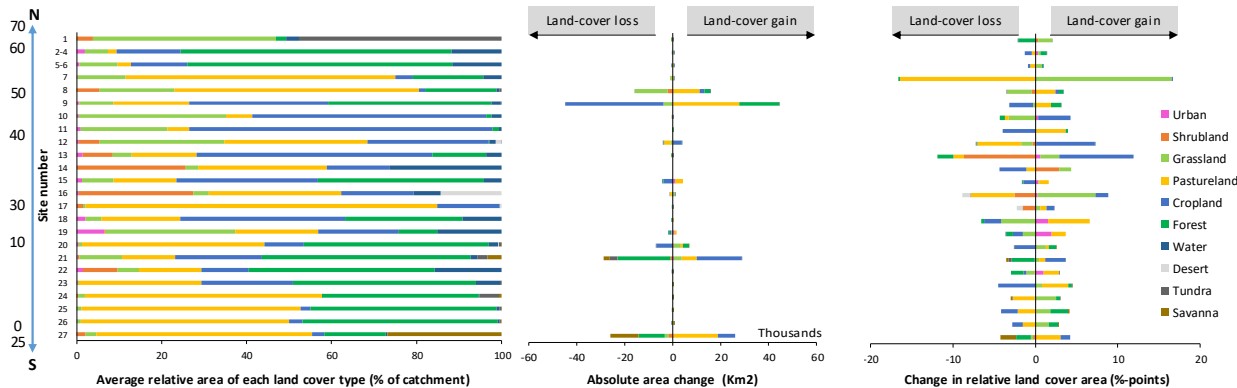

**Figure 7.** (Left) Long-term average relative area of each land cover type (percentage of total wetlandscape area). (Center) Absolute change in area of each land cover type (km²). (Right) Change in relative land cover area (%-points in relation to total catchment area). The summarized and illustrated data are for the 27 wetlandscapes included in WetCID.

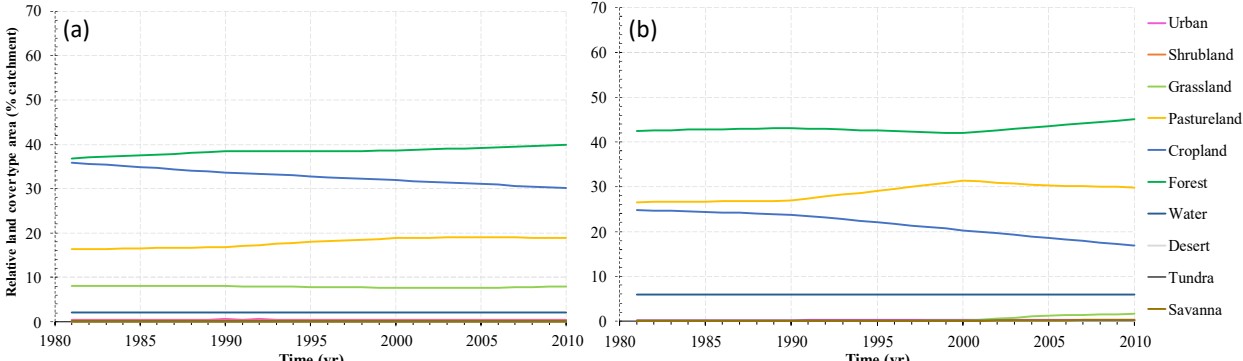

**Figure 8.** Data time series for wetlanscape-aggregated annual average area (relative to total wetlandscape area, in %) for different land cover types in the (a) Volga and (b) León-Atrato wetlandscapes.

The data time series of different land covers and their changes between Per1 (1981-1995) and Per2 (1996-2010) show, for example, forest and (decreasing) cropland, followed by pastureland and grassland, to be dominant in the large Volga wetlandscape, while forest, pastureland and (decreasing) cropland areas dominate the small León-Atrato wetlandscape (Figure 8). Gridded maps of land cover area changes in these wetlandscape examples (Figures 9-10) again demonstrate large spatial resolution differences with potentially important implications for the usefulness of land use datasets for wetlandscapes of smaler scale. For example, in the most northern Swedish-Arctic wetlandscape

1, grassland is obtained as the second dominant landcover type after tundra (Figure 7, left plot), which is not normally
seen in this northern Arctic region.

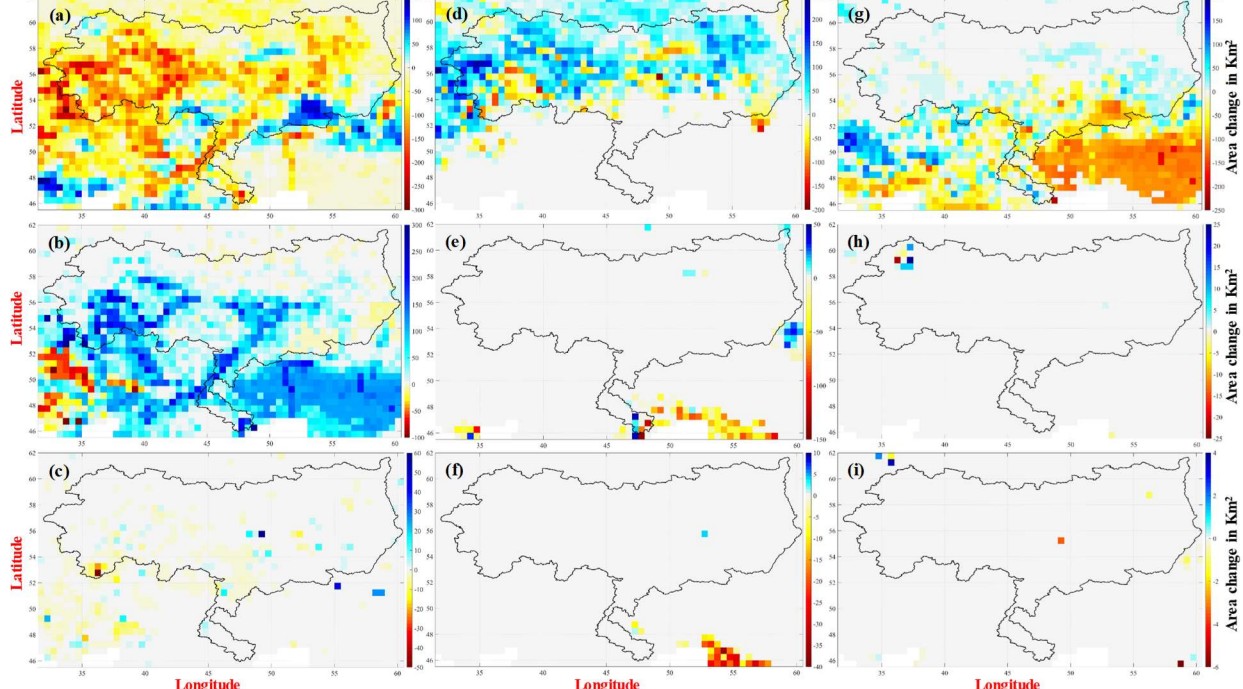

**Figure 9.** Gridded maps of absolute area changes (in km$^2$) for (a) cropland, (b) pasturland, (c) urban, (d) forest, (e)
shrubland, (f) desert, (g) grassland, (h) tundra, and (i) water land cover types between Per1 (1981-1995) and Per2
(1996-2010) in the Volga wetlandscape example.

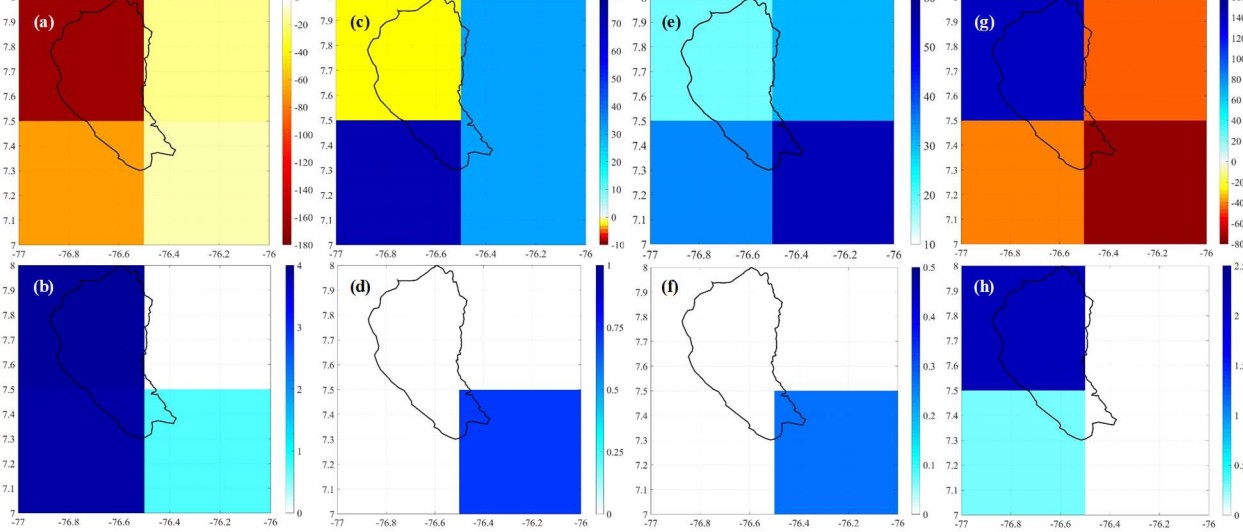

**Figure 10.** Gridded maps of absolute area changes (in km$^2$) for (a) cropland, (b) savanna, (c) forest, (d) shrubland, (e)
grassland, (f) tundra, (g) pasturland, and (h) urban land cover types between Per1 (1981-1995) and Per2 (1996-2010)
in the León-Atrato wetlandscape example.

**4 Data availability**

The complete WetCID database includes five file categories (https://doi.pangaea.de/10.1594/PANGAEA.907398; Ghajarnia et al., 2019).

*Folder 1: Survey results (Summary documents A, B, C)*

These three summary documents (all in Excel) were created from responses obtained in the main survey of GWEN researchers (see survey template/structure in WetCID files). Summary document A contains summarized site-specific information on the wetlands, hydrology, climate and land uses in each of for the 27 wetlandscapes. Summary documents B and C contain local knowledge relating to the availability-accessibility (or lack) of land use and hydroclimatic data, respectively, for each of the 27 wetlandscapes.

*Folder 2: Gridded land use and hydroclimatic datasets (NetCDF database files)*

In WetCID, there is a separate NetCDF file for each wetlandscape that contains a complete set of gridded hydroclimate and land use data time series for the closest rectangular window around the catchment polygon of the wetlandscape. The gridded hydroclimate datasets were created by subsetting the CRU_TS4.02 original global datasets over the area of each wetlandscape (catchment). The gridded land use dataset for each wetlandscape (catchment) was created by first reclassifying the land cover types and then subsetting the global gridded data. All these gridded data time series are saved in separate NetCDF files for each wetlandscape, which is an appropriate file type for storing gridded data. Each NetCDF file contains 18 variables, including hydroclimate, land cover, and some auxiliary variables. Appendix B presents the general attributes table (Table B1) and information and explanations of all 18 variables included in the NetCDF database files (Table B2). Sample Matlab and R codes for reading and extracting data from the NetCDF files are also provided in Appendix C.

*Folder 3: Aggregated land use and hydroclimate data (Excel databases)*

The time series of land use and hydroclimatic data aggregated over each wetlandscape (catchment) were created from the gridded datasets (NetCDF files) and stored as Excel files for each wetlandscape. The Excel file for each wetlandscape contains three sheets: 1) Annual time series of covered area by each land cover type in $km^2$, 2) time series of annual relative area (%) occupied by each land cover type, and 3) time series of monthly temperature (°C) and precipitation (mm/month) data.

*Folder 4: Geographical dataset in a zip file (shapefiles)*

To perform any spatial analysis of the wetlandscapes, one needs to have access to the shapefile and polygons of the wetlandscape (catchment) and wetlands within it. These shapefiles were provided by the GWEN researchers and can be downloaded from WetCID files.

*Folder 5: Summary tables of changes in hydroclimatic and land use variables*

Absolute and relative changes in all considered hydroclimate and land use variables between Per1 (1981-1995) and Per2 (1996-2010) were calculated using Eq. (1), (2), and (3) for each wetlandscape. The results are summarized in an Excel file with two sheets for each wetlanscape: 1) Absolute changes in temperature, precipitation and land cover area, and 2) relative changes in precipitation and land cover area. The data for land cover changes are provided for all considered land use variables.

**5 Conclusions**

The presented new database (WetCID) combines survey-based local information and knowledge with gridded large-scale hydroclimate and land use datasets for 27 wetlandscapes around the world. The gridded datasets contain 30-year time series of mean monthly precipitation and temperature, along with annual average land uses and their changes over this time period for each wetlandscape. WetCID can support site assessments, cross-regional comparisons, and scenario analyses of the roles and impacts of various land use, hydroclimatic and wetland conditions and their changes

on whole-wetlandscape functions and associated ecosystem services. The information on local data availability/accessibility and observed/perceived change occurrence summarised and structured in WetCID can guide further study directions and support identification of key needs for complementary new local data and/or use of additional regional-global gridded datasets.

The gridded large-scale hydroclimatic and land use data included in WetCID have been derived using open data sources and processed with open-source tools, while WetCID has been designed so that more data can readily be added to it. The site-specific usefulness of different included data varies for wetlandscapes of different scales, but WetCID can be updated with small time investment as new datasets become available, or current datasets are expanded or refined.

## Acknowledgements

This study was supported by funding from Swedish Research Council Formas (grant number 2016-2045). The Historical Land cover Change and Land use Conversions Global Dataset used in this study was acquired from NOAA's National Climatic Data Center (http://www.ncdc.noaa.gov/). The temperature and precipitation data was also retrived from the CRU_TS4.02 global database (https://crudata.uea.ac.uk/cru/data/hrg/). The data of Selenga and Volga wetlandscapes were prepared within RFBR project 17-29-05027 and 18-05-60219. Travel to the workshop was made possible for some authors with support from the National Science Foundation trhough the Florida Coastal Everglades Long-Term Ecological Research Program under Grant No. DEB-1237517 (contribution number XXX from the Southeast Environmental Research Center at Florida International University).

## Author contributions

N.G. compiled the climate and land use database, contributed to the communication with other co-authors for the wetlandscape data collection, and was main responsible for analyzing the data and writing the paper. G.D. conceived and led the study and the development of WetCID and analysis approach, led the communication with other co-authors, and contributed to the result analysis and writing of the paper. J.T. conceived the idea of the data paper type, was main responsible for collecting and compiling the local survey information and its summary and analysis in the paper, and contributed to communication with co-authors, the result analysis and the writing. Z.K. contributed to the communication with co-authors, the database development, and the result analysis and writing. All other co-authors contributed by providing local site information in the survey forms and/or taking part in discussions for planning and outlining the study.

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

## Appendix A: Summary of land cover type parameters

**Table A1.** List of all different land cover types included in the NOAA-HYDE dataset and their corresponding reclassified category in WetCID

| Number | Land Cover Name | Description | Reclassified Category |
|---|---|---|---|
| 1 | TrpEBF | Tropical Evergreen Broadleaf Forest | Forest |
| 2 | TrpDBF | Tropical Deciduous Broadleaf Forest | Forest |
| 3 | TmpEBF | Temperate Evergreen Broadleaf Forest | Forest |
| 4 | TmpENF | Temperate Evergreen Needleleaf Forest | Forest |
| 5 | TmpDBF | Temperate Deciduous Broadleaf Forest | Forest |
| 6 | BorENF | Boreal Evergreen Needleleaf Forest | Forest |
| 7 | BorDNF | Boreal Deciduous Needleleaf Forest | Forest |
| 8 | Savannah | Savannah | Savannah |
| 9 | C3grass | C3 Grassland/Steppe | Grassland |
| 10 | C4grass | C4 Grassland/Steppe | Grassland |
| 11 | Denseshrub | Dense Shrubland | Shrubland |
| 12 | Openshrub | Open Shrubland | Shrubland |
| 13 | Tundra | Tundra | Tundra |
| 14 | Desert | Desert | Desert |
| 15 | PdRI | Polar Desert/Rock/Ice | Desert |
| 16 | SecTrpEBF | Secondary Tropical Evergreen Broadleaf Forest | Forest |
| 17 | SecTrpDBF | Secondary Tropical Deciduous Broadleaf Forest | Forest |
| 18 | SecTmpEBF | Secondary Temperate Evergreen Broadleaf Forest | Forest |
| 19 | SecTmpENF | Secondary Temperate Evergreen Needleleaf Forest | Forest |
| 20 | SecTmpDBF | Secondary Temperate Deciduous Broadleaf Forest | Forest |
| 21 | SecBorENF | Secondary Boreal Evergreen Needleleaf Forest | Forest |
| 22 | SecBorDNF | Secondary Boreal Deciduous Needleleaf Forest | Forest |
| 23 | Water | Water/Rivers | Water |
| 24 | C3crop | C3 Cropland | Cropland |
| 25 | C4crop | C4 Cropland | Cropland |
| 26 | C3past | C3 Pastureland | Pastureland |
| 27 | C4past | C4 Pastureland | Pastureland |
| 28 | Urban | Urban land | Urban |

**Appendix B: Description of parameters included in the NetCDF database files of WetCID**

**Table B1.** General attributes table for NetCDF database files of WetCID

| Item | Description |
|---|---|
| project_name | Global Wetland Ecohydrology Network (GWEN) – An Agora for Scientists and Study Sites |
| project_summary | GWEN consists of a network of wetland researchers at study sites around the world, who are all interested in sharing, investigating, and applying research to improve knowledge on the large-scale function of, and changes to, wetland ecosystems. |
| project_website | http://www.gwennetwork.se/ |
| dataset | land use and climate data for the catchments of wetlands included in GWEN |
| comment | The dataset in this NetCDF file is created to represent the change in land use and land cover over the catchment area of each wetland site included in the GWEN project. Precipitation and temperature time series data are also included for climate considerations. |
| land use data_reference | NOAA-Historical Land-Cover Change and Land-Use Conversions Global Dataset_HYDE version (https://data.nodc.noaa.gov/cgi-bin/iso?id=gov.noaa.ncdc:C00814) |
| climate data_reference | Climate Research Unit (CRU) data CRU_TS v. 4.02 (https://crudata.uea.ac.uk/cru/data/hrg/cru_ts_4.02/) |
| license | please quote the following citation when using data: .... |
| data_type | grid |
| spatial_resolution | 0.5x0.5 degrees latitude/longitude |
| institution | Dept. of Physical Geography, Stockholm University, Sweden |
| time_coverage_start | 1981 |
| time_coverage_end | 2010 |
| time_coverage_resolution | yearly for land cover data and monthly for climate data |
| date_created | May-19 |
| core group of researchers determining the dataset | Georgia Destouni, Navid Ghajarnia, Zahra Kalantari, Josefin Thorslund |
| creator name | Navid Ghajarnia |

**Table B2.** List and description of land use and hydroclimate variables included in the NetCDF database files of
488 WetCID

| Number | Variable Name | Variable Long Name | Variable Explanation |
|---|---|---|---|
| 1 | longitude | longitude | degrees_east |
| 2 | latitude | latitude | degrees_north |
| 3 | time_LCD | time for land cover datasets | years since, 1 January 0001 |
| 4 | time_CD | time for climate datasets | days since 1900-1-1 |
| 5 | Mask | Grids that have/have not overlap with catchment area | catchment area binary mask [0,1] |
| 6 | Area | Area of land grid cells | Units are in $km^2$ |
| 7 | Urban | Urban land cover type | Units are in percentage of grid cell area |
| 8 | Shrubland | Open/dense shrubland land cover type | Units are in percentage of grid cell area |
| 9 | Grassland | Grassland/steppe land cover type | Units are in percentage of grid cell area |
| 10 | Pastureland | Pastureland land cover type | Units are in percentage of grid cell area |
| 11 | Cropland | Cropland land cover type | Units are in percentage of grid cell area |
| 12 | Forest | Tropical, Temperate, Boreal Evergreen, Deciduous Broadleaf, Needleleaf Forest land cover type | Units are in percentage of grid cell area |
| 13 | Water | Water/rivers land cover type | Units are in percentage of grid cell area |
| 14 | Desert | Desert/polar desert/rock/ice land cover type | Units are in percentage of grid cell area |
| 15 | Tundra | Tundra land cover type | Units are in percentage of grid cell area |
| 16 | Savannah | Savannah land cover type | Units are in percentage of grid cell area |
| 17 | Prcp | Precipitation | Units are in mm/month |
| 18 | Tmp | Near-surface temperature | Units are in degrees Celsius |

489

**Appendix C: Sample codes to read NetCDF database files included in WetCID**

*Matlab Sample code:*

info = ncinfo('File_Name.nc'); % replace File_Name with the name of NetCDF file for each wetlandscape. This command gets the complete description for all the general attributes as well as detailed information of all existing variables in the NetCDf file.

Var = ncread('File_Name.nc', 'Variable_Name'); % replace Variable_Name with the Variable Name column in Table B2 for extracting different variable data from each wetlandscape NetCDF file.

*R Sample code:*

install.packages("ncdf4")

library(ncdf4)

ncf <- nc_open("File_Name.nc ") # replace File_Name with the name of NetCDF file for each wetlandscape. This command opens the NetCDF file in RStudio environment.

names(ncf$var) # extracting the name of existing variables in the NetCDF file.

Var <- ncvar_get(ncf, " Variable_Name ") # replace Variable_Name with the Variable Name column in Table B2 for extracting different variable data from each wetlandscape NetCDF file.