# Peer review of "Data for wetlandscapes and their changes around the world 1"

_Earth System Science Data, 2019_

## Referee Comment (RC1) · Anonymous Referee #1 · 1 Mar 2020

The manuscript is generally very important because it summarizes the state of wetlands worldwide. But, though it is not the intention of the manuscript, the question is: Could it help to manage wetlands at a regional scale?

---

## Referee Comment (RC2) · Anonymous Referee #2 · 5 Mar 2020

Proposed manuscript presents the database of selected wetlandscapes around the world. The novelty of the database proposed by Navid Ghajarnia et al. is the time series of climate and landuse data. Existing GIS databases related to wetlands include the wetland boundaries only and some only the name, area and protection status. Proposed database can be used for analysis of wetlands changes. The database is characterized by usability because the possibility of adding the new objects. More-over, the idea of using "the wetlandscapes" underlines that wetlands are component of landscape. This component is in close interaction with climate, water bodies, landuse etc. Proposed manuscript presents in clear way the idea of database, used meth-ods, sources of data and show possible use of the database for analysis of wetland-scapes changes in regional and global scale. Slight weaknesses were noticed in the

manuscript. I recommend the Authors to add to the manuscript the table with basic information of wetlandscapes showed in Figure 1. Uniform black dots used for showing the presented individual objects on the map suggest that these objects are of similar size. However, based on the data presented in "Survey summary doc A_General site information" show that presented wetlandscapes are in differ total area and percentage of wetlands. This information added in table into manuscript could provide this basic information for readers. In my opinion, key information is also altitude (mean in case of small objects and mean, maximum and minimum in case of large ones) and salinity. Moreover, small technical correction in "Survey summary doc A_General site information – site info" are recommended: unify of font and some detail information, eg. in some cases there is information "326102 number of people for watershed", however in the others "940 hab. (2002)", "Total population of 2.9 million people implies an average population density of $\sim$ 78 people per km2". It would be better to unify the way of presentation the population data, lakes data etc.

---

## Author Comment (AC2) · 17 Mar 2020

Suggestions by Anonymous Referee #2: I recommend the Authors to add to the manuscript the table with basic information of wetlandscapes showed in Figure 1. Uniform black dots used for showing the presented individual objects on the map suggest that these objects are of similar size. However, based on the data presented in "Survey summary doc A_General site information" show that presented wetlandscapes are in differ total area and percentage of wetlands. This information added in table into manuscript could provide this basic information for readers. In my opinion, key information is also altitude (mean in case of small objects and mean, maximum and minimum in case of large ones) and salinity. Moreover, small technical correction in "Survey summary doc A_General site information – site info" are recommended: unify of font

and some detail information, eg. in some cases there is information "326102 number of people for watershed", however in the others "940 hab. (2002)", "Total population of 2.9 million people implies an average population density of _ 78 people per km2". It would be better to unify the way of presentation the population data, lakes data etc.

Response: Thanks for your feedback and suggestions. Based on your suggestion, we have now added Table 1 in section 3.1 of the revised manuscript. This table presents some selected fields of the "summary table sheet" in the "Survey summary doc A", including site names, country, classification (wetlandscape or individual wetlands), climate zone, wetland type, and area of wetlands relative to total wetlandscape (catchment) area. The total wetlandscape (catchment) area is not added in Table 1 as this information is embedded as a graph in Figure 2. Specific salinity levels are not included either, as such data have not been provided for the different sites, and the focus of the current version of the Wetlandscape Change Information Database (WetCID) is on land use and hydroclimatic changes, rather than on water quality. However, the wetland type for each site, which is given in Table 1, clarifies if there are mainly freshwater, brackish water, or saline water wetlands at each site. Various water quality data are definitely interesting to consider and include in further developments of the database. Moreover, altitude is also useful information to add in future developed versions of WetCID database, even though it has not been collected and prepared at this stage. The fonts, size and general format cells of Survey summary doc A are revised based on your recommendations (see the attached file). Different information items regarding latitude, longitude, climate zones, temperature, evapotranspiration, runoff, precipitation, groundwater table, and population, are also harmonized as much as possible. However, please note that, although we have defined a general structure for different existing fields in the survey forms (to harmonize reported data), we tend to keep additional information provided by each site's researchers, as this provides valuable local insights. Therefore, this may cause some minor inconsistencies in level of available data among different sites. The added Table 1 and its associated explanation in the revised manuscript are as follows: In the beginning of section 3.1,
between lines 188-191: "Table 1 summarizes some general geographical, climate, and wetland type information provided by GWEN researchers in the survey information forms. Each site represents either an individual wetland or a wetlandscape (e.g., a catchment) including multiple wetlands. The country, main climate zone and wetland area relative to total wetlandscape (catchment) area are also given for each site in Table 1."

Please also note the supplement to this comment:
https://www.earth-syst-sci-data-discuss.net/essd-2019-207/essd-2019-207-AC2-supplement.zip
* * *
1     **Table 1.** General geographic, climate, and wetland type information for the 27 investigated wetlandscapes in WetCID. The data and information are based on
2     survey responses by researchers with active research (on various topics) at each wetlandscape site.

| Site No. | Site name | Country | Classification | Climate zone | Wetland type | Area of wetlands relative to total catchment/wet-landscape area (%) |
|---|---|---|---|---|---|---|
| 1 | Tavvavouma | Sweden | Wetlandscape | Subarctic | Peat plateau/thermokarst lake complex | 2.8 |
| 2 | Forsmark | Sweden | Wetlandscape | Humid continental (cold summer) | Bogs, fens, marshes, (shallow lakes) | 0.01 |
| 3 | Vattholma | Sweden | Wetlandscape | Humid continental (cold summer) | Bog, Fen, Riparian | - |
| 4 | North Baltic WMD | Sweden | Wetlandscape | Humid continental (cold summer) | Multiple | 100 |
| 5 | Simpevarp | Sweden | Wetlandscape | Humid continental (cold summer) | Bogs, fens | 0.01 |
| 6 | South Baltic WMD | Sweden | Wetlandscape | Humid continental (cold summer) | Multiple | 100 |
| 7 | Upper Lough Erne | Ireland | Individual wetland | Cold (dry winter, cold summer) | Flood plain/shallow lakes | 22 |
| 8 | Selenga | Russia | Wetlandscape | Cold (dry winter, cold summer) | Marshes (Riverine, Palustrine) | 0.13 |
| 9 | Volga | Russia | Wetlandscape | Cold (dry winter, cold summer) | Marshes (Riverine, Palustrine) | 1.0 |
| 10 | Le Sueur | USA | Wetlandscape | Temperate | isolated, fluvial/riparian, lakes/ponds, marshes, forest/shrubs, constructed | 100 |
| 11 | Sacca Di Goro | Italy | Individual wetland | Cold-summer Mediterranean | Shallow saltwater coastal lagoon | 4.2 |
| 12 | Lake Urmia | Iran | Individual wetland | Continental | Lake | 8.8 |
| 13 | Anzali Mordab | Iran | Individual wetland | Caspian or Hyrcanian climate | Inland and Marine/Coastal wetland | 4.0 |
| 14 | Gialova Lagoon | Greece | Individual wetland | Hot-summer Mediterranean | Coastal wetland | 13 |
| 15 | Lower Mississippi River Delta Plain | USA | Wetlandscape | Humid Subtropical | Riverine, Marine, Estuarine, Lacustrine | 3.5 |
| 16 | Shadegan | Iran | Individual wetland | Warm desert | Palustrine, Estuarine, Marin | 31 |
| 17 | Zone Humide de Souss | Morocco | Individual wetland | Mediterranean semi-arid | marine and coastal | 0.01 |
| 18 | Geographically isolated wetlands | USA | Wetlandscape | Humid subtropical | Freshwater marshes and swamps | 100 |
| 19 | Everglades | USA | Individual wetland | Tropical to Subtropical | Freshwater wetland, coastal wetland | 32 |
| 20 | CGSM | Colombia | Individual wetland | Tropical | Estuarine | - |
| 21 | Mekong Delta | Vietnam | Wetlandscape | Tropical Monsoon | Marine | 5.0 |
| 22 | Panama Canal | Panama | Wetlandscape | Tropical/Central America | River Chagres, Lake | 100 |
| 23 | León-Atrato | Colombia | Wetlandscape | Tropical rainforest | Marshes and Swamps | 17 |
| 24 | Lagunas Plaza and Grande | Colombia | Wetlandscape | Extremely cold and very dry | Glacial Lake | 4.4 |
| 25 | Fúquene, Cucunubá y Palacio | Colombia | Individual wetland | Cold and very dry | Natural shallow lake | 1.7 |
| 26 | Paramo Sumapaz | Colombia | Wetlandscape | Tropical | High altitude wetland | 46 |
| 27 | Pantanal | Brazil | Wetlandscape | Tropical savanna with dry-winter | Periodically inundated savanna | 27 |

**Fig. 1.** Table 1. General geographic, climate, and wetland type information for the 27 investigated wetlandscapes in WetCID

---

## Author Response (AR1)

**Dear Editor of *Earth System Science Data***

We hereby submit the revised version of our manuscript entitled "Data for wetlandscapes and their changes around the world" in response to the interactive review process in the ESSD journal. The revised manuscript and response files have been prepared by the core group of authors, including Navid Ghajarnia, Georgia Destouni, Josefin Thorslund, and Zahra Kalantari, and finally reviewed and revised by all other co-authors listed in the manuscript.

We thank the editor and reviewers for their constructive comments and suggestions. The paper is revised based on the comments given by the reviewers as explained in the "*Response file and manuscript revised_Changes highlighted*" file, with changes highlighted in red. Detailed responses to each and all of the comments and questions of both reviewers are also provided in the same file. All co-authors agree with the submitted revised version of the manuscript.

On behalf of me and all the co-authors
Yours sincerely,
Navid Ghajarnia

**Question by Anonymous Referee #1:**

The manuscript is generally very important because it summarizes the state of wetlands worldwide. But, though it is not the intention of the manuscript, the question is: Could it help to manage wetlands at a regional scale?

**Response:**

Thanks for your feedback.

The main idea in this research is to support and encourage large-scale studies of geographical, hydrological, hydroclimate and land use conditions and changes over the whole wetlandscapes, at different sites around the world that enables cross-regional comparisons. The introduced sites in the wetlandscape change database (WetCID) are located in different regions, with seven sites in Northern Europe (sites 1-7), seven in the Amazon and Caribbean region (sites 20 and 23-27), four in North America (sites 10, 15, 18, and 19), three in the Middle East (sites 12, 13, and 16), two in the Mediterranean region (sites 11 and 14), two in Siberia (sites 8 and 9), and two more in other parts of the world (Northern Africa and East Asia). As such, regional change patterns and characteristics can be identified, specifically over regions with higher number of sites. It can also highlight regional knowledge gaps or data availability as critical obstacles towards proper management of wetlands which is a first step to fulfill such shortcomings (e.g. lack of open data sources identifies for sites 13, 16, and 17 in the MENA region reported in lines 192-193 or lack of information regarding observed changes at sites 2 and 5 in North Europe reported in lines 198-199). Moreover, given the fact that this is a work in progress and the database can be updated with small time investment as information from new sites become available, more comprehensive regional evaluations can be expected from future versions of the WetCID.

In order to address this comment and clarify about it in the manuscript for similar interested audiences and WetCID users, the following parts of the manuscript are revised while lines 54-55 and 347-349 already discuss around the above question in the submitted version.

After the paragraph in lines 195-202 and before Figure 2, the following sentences are added: "Furthermore, the wetlandscapes of WetCID are located in different regions of the world, with seven sites in Northern Europe (sites 1-7), seven in the Amazon and Caribbean region (sites 20 and 23-27), four in North America (sites 10, 15, 18, and 19), three in the Middle East (sites 12, 13, and 16), two in the Mediterranean region (sites 11 and 14), two in Siberia (sites 8 and 9), and two more in other parts of the world (Northern Africa and East Asia). As such, regional patterns and characteristics can be identified, and regional strategies developed, e.g., to enhance availability of data and information, and determine further research needed to bridge region-specific knowledge gaps and decide on relevant management plans for each region's wetland ecosystems. Such regional characterizations and assessments can be initialized with the current version of WetCID and further updated as more data for already included and possible additional regional wetlandscapes become available in future database versions."

**Suggestions by Anonymous Referee #2:**

I recommend the Authors to add to the manuscript the table with basic information of wetlandscapes showed in Figure 1. Uniform black dots used for showing the presented individual objects on the map suggest that these objects are of similar size. However, based on the data presented in "Survey summary doc A_General site information" show that presented wetlandscapes are in differ total area and percentage of wetlands. This information added in table into manuscript could provide this basic information for readers. In my opinion, key information is also altitude (mean in case of small objects and mean, maximum and minimum in case of large ones) and salinity. Moreover, small technical correction in "Survey summary doc A_General site information – site info" are recommended: unify of font and some detail information, eg. in some cases there is information "326102 number of people for watershed", however in the others "940 hab. (2002)", "Total population of 2.9 million people implies an average population density of _ 78 people per km2". It would be better to unify the way of presentation the population data, lakes data etc.

**Response:**

Thanks for your feedback and suggestions.

Based on your suggestion, we have now added Table 1 in section 3.1 of the revised manuscript. This table presents some selected fields of the "summary table sheet" in the "Survey summary doc A", including site names, country, classification (wetlandscape or individual wetlands), climate zone, wetland type, and area of wetlands relative to total wetlandscape (catchment) area. The total wetlandscape (catchment) area is not added in Table 1 as this information is embedded as a graph in Figure 2. Specific salinity levels are not included either, as such data have not been provided for the different sites, and the focus of the current version of the Wetlandscape Change Information Database (WetCID) is on land use and hydroclimatic changes, rather than on water quality. However, the wetland type for each site, which is given in Table 1, clarifies if there are mainly freshwater, brackish water, or saline water wetlands at each site. Various water quality data are definitely interesting to consider and include in further developments of the database. Moreover, altitude is also useful information to add in future developed versions of WetCID database, even though it has not been collected and prepared at this stage.

The fonts, size and general format cells of Survey summary doc A are revised based on your recommendations (see the attached file). Different information items regarding latitude, longitude, climate zones, temperature, evapotranspiration, runoff, precipitation, groundwater table, and population, are also harmonized as much as possible. However, please note that, although we have defined a general structure for different existing fields in the survey forms (to harmonize reported data), we tend to keep additional information provided by each site's researchers, as this provides valuable local insights. Therefore, this may cause some minor inconsistencies in level of available data among different sites.

The added Table 1 and its associated explanation in the revised manuscript are as follows:

[revised manuscript text omitted]